# Understanding Healthcare Professionals’ Knowledge on Perinatal Depression among Women in a Tertiary Hospital in Ghana: A Qualitative Study

**DOI:** 10.3390/ijerph192315960

**Published:** 2022-11-30

**Authors:** Sandra Fremah Asare, Maria F. Rodriguez-Muñoz

**Affiliations:** 1Department of Personality, Assessment and Psychological Treatment, Universidad Nacional de Educación a Distancia, 28015 Madrid, Spain; 2Discipline of Nursing, School of Nursing and Public Health, College of Health Sciences, University of KwaZulu-Natal, Durban 4001, South Africa; 3Seventh-Day Adventist Nursing and Midwifery Training College, Kwadaso, Kumasi P.O. Box PC 96, Ghana

**Keywords:** health professionals, Ghana, perinatal depression, knowledge, qualitative study

## Abstract

Health conditions affecting women in the perinatal period still account for a major contribution to disease burden in Sub-Saharan Africa, yet there is a dearth of empirical research to understanding health professionals’ perspective on their experiences and how they care for perinatal women in depression. We used a qualitative exploratory descriptive approach through a face- to face-interview to explore the knowledge of 11 health professionals of Komfo Anokye Teaching Hospital, Kumasi- Ghana. Interviews were taped recorded and transcribed verbatim. The study adopted Haase’s modification of Colaizzi’s method for the analysis. Four main themes emerged: ineffective communication (Referral lapses among care providers, long waiting time, lack of confidentiality), workload (Inefficient staff to meet perinatal women’s need, no screening tools and time constraints), Reaction to patients symptoms (Identifying patient’s symptoms, assessment through patient’s centeredness and Education and counselling). Our results emerged that time constraints, stigmatization and lack of awareness delayed the care and management of perinatal depression among healthcare professionals in the hospital setting. There is the need to improve healthcare professionals’ knowledge on perinatal depression and it is imperative for the hospital administrators to invest in continuous training and professional development for healthcare professionals.

## 1. Introduction

Perinatal depression (PD) is a major depressive disorder during pregnancy or within 4 weeks after childbirth up to a year [1], and it is a public health concern. Although the most recent multidisciplinary research on PD has focused on risk factors, effects, treatment and prevention, global knowledge is still lacking on PD [2]. Evidence suggests that the most common psychological problem that affect women in the perinatal period is depression [3,4]. Global disparities in population mental health and mental health systems are paralleled by disparities in the evidence base supporting effective intervention for PD [5]. According to the World Health Organization, 1 in 10 women develop perinatal depression in High-income Countries whereas, in Lower Middle-income Countries, 1 in 5 women suffer from perinatal depression [6]. PD is common and untreated PD is associated with inimical effects on the mother, fetus, child and family [7]. Timely, appropriate and effective care for women with prenatal and postnatal depression is required of healthcare professionals (HCPs) for better outcomes [8]. Research has shown that health conditions affecting the perinatal period still account for a major contribution to disease burden in Sub-Saharan Africa (SSA) [9], yet little research exist on HCPs’ knowledge on perinatal mental health-related issues. PD is noted to have long-term consequences and especially depression in the perinatal period diminishes a woman’s capacity for self-care and dependence, and further affects a woman’s physical and mental health [10]. Various researchers from SSA have found that various causes of prenatal depression such as financial difficulties, relationship problems such as polygamy, lack of support, neglect and infidelity, loss of previous pregnancy and even fear of birth outcomes, and traumatic events, that is intimate partner violence and excessive alcohol use by the husband were determinants of postnatal depressive symptoms [11,12,13]. Notably, studies have indicated the positive role of fathers in maternal healthcare [14,15], and perinatal women who enjoy the support of their husbands feel empowered and are able to cope with the stress and difficulties associated with pregnancy and childbirth [14]. Thus, there is substantial evidence that husbands’ participation has positive effects such as decreases in the risk of preterm babies, low birth weight, stunted postpartum growth in newborns, and infant mortality [16,17].

Multiple authors have found the prevalence of suicidal ideations among pregnant women to be higher in SSA and in low-income women [18,19,20]. Evidence from LMICs indicate that there are higher rates of suicidal ideation in perinatal women compared with high-income contexts that range from 14.0 to 27.5% [21,22]. A recent study has shown that perinatal women who conceive suicidal ideations are much more likely to have chronic comorbid depressive and anxiety disorders than non-pregnant women [2]. Research has found that managing women with perinatal depression is well-known among midwives in high-income countries (HIC) [8,23]. In LMICs however, stigmatization and a lack of awareness of the care and management of perinatal depression among HCPs have been reported [24,25]. A substantial number of studies have revealed that perinatal women have been subjected to disrespectful and abusive care by HCPs, especially during childbirth [26,27,28]. A recent systematic review by Minckas et al. also affirmed the mistreatment and disrespectful care from healthcare professionals to women during childbirth [29]. Perinatal depression, although being one of the most severe obstetric complications, have a dearth of evidence of literature on HCPs knowledge of perinatal depression especially in Lower and Middle-Income countries (LMIC) including Ghana. Research in Sub-Saharan Africa have found prevalence of depression in mothers. In Nigeria, perinatal depression among women was reported to be 10–30% [30,31] and in Ethiopia 18.28%, using the Edinburg Postnatal Depression Scale (EPDS), [32], and in Uganda, 6.1% during the post-partum period [33]. The EPDS, though not a diagnostic tool, helps detect the level of symptoms and risk of depression among perinatal women [34]. Evidently, a study showed that there is a lack of understanding of the woman’s need for support during and after hospital stay and the need for clear information by healthcare professionals [35].

In Ghana, a recent study conducted in three public hospitals concurrently, namely a primary health facility, a secondary facility, and a tertiary facility using the EPDS found that the prevalence of PD was 8.6%, 31.6%, and 41.1%, respectively [36]. Although the regular contact between mothers and perinatal HCP may make the obstetric setting ideal for addressing depression, data suggests that there are existence of barriers, and PD remains under-diagnosed and under-treated [37]. The National Institute for Health and Care Excellence (2018) has recommended routine screening for women due to risk for mental health problems in the perinatal period [38]. However, empirical evidence suggests that in low-resource settings, screening instruments for depression may be absent or limited, as they may be considered too long and time-consuming for routine screening in busy antenatal clinics [39,40], and therefore, depression may go unnoticed or undertreated.

Policies aimed at identifying perinatal women with a history of depression by healthcare professionals are necessary for early intervention and may well abate the potentially negative consequences of perinatal depression for maternal-child health [41]. Despite the outcome of perinatal depression and its deleterious effect on mother and child, there is a dearth of evidence on qualitative research on HCPs’ knowledge on perinatal depression, especially in Ghana. This background necessitates the need for studies to determine the factors that can be improved to promote maternal mental health. The present study is aimed at exploring the knowledge of HCPs on perinatal depression among women in a tertiary hospital in Kumasi, Ghana.

## 2. Materials and Methods

### 2.1. Design

An empirical phenomenological approach was used to obtain detailed experiences from the HCPs [42]. Twelve professionals were purposefully asked to participate and one opted out. Data saturation was reached with eleven participants. All three doctors were males and the eight females were midwives. The study used a qualitative exploratory descriptive approach to answer the research question: what knowledge do expert healthcare professionals have in recognizing and managing perinatal mental health issues especially depression in women? An open-ended questionnaire which served as the interview guide was developed and allowed the HCPs to open up about their knowledge of perinatal mental healthcare. In adopting this method, the knowledge gap among HCPs about perinatal mental health issues was explored, understood, and described [43]. This design was useful as little is known about HCPs knowledge on perinatal mental health in Ghana [44]. COREQ checklist for reporting qualitative research guided the study [45].

### 2.2. Study Setting

The study was conducted at the Komfo Anokye Teaching Hospital (KATH) in Kumasi. It is a referral facility and one of the largest teaching hospitals located in the Ashanti Region of Ghana, serving patients across the country and neighboring West African countries and has a bed capacity of over one thousand, two hundred (1200). It has 12 clinical directorates and several units including the specialist antenatal and post-natal care clinics as well as the emergency unit of the Polyclinic which attends to all obstetrics and gynecological emergency cases on a 24-h basis and served as the main recruitment outlet. The key function of the directorate include the provision of general and specialist women’s healthcare. In 2020, twelve thousand, eight hundred and twenty-seven (12,827) women were seen for antenatal care services whilst four thousand and ninety-three (4093) women sought for post-natal care at the Obstetrics and Gynecological (O&G) Directorate [46].

### 2.3. Population and Sampling Technique

In all eleven HCPS who were purposively sampled, participated in the study. These consisted of an obstetrician-gynecologist and midwives who dealt directly with pregnant and postpartum women and psychiatrists at the hospital constituted the study population. The inclusion criteria included HCPs working at the antenatal and postnatal clinics and the emergency out-patient department of O&G, consisting of obstetrician-gynecologist, midwives and psychiatrists who attend to perinatal women in depression were enrolled and have worked at the various department for at least 3 years and above. Purposive sampling was to ensure that health professionals who had expert knowledge on maternal mental health matters were recruited into the study to share their experiences on the phenomenon of interest [44]. Prior to the study, the first author approached the midwives and doctors individually, explained the study objectives and obtained verbal and written consent. Three interviews were conducted each week to allow for transcription and coding to ascertain emerging themes [26]. For recall bias to reduce, healthcare professionals recruited for the study worked at different units within the O&G Directorate and the OPD. Saturation of data was achieved with eleven (11) health workers with no new themes emerging [47].

### 2.4. Data Collection Process

A total number of 11 face-to-face in-depth individual interviews were conducted. Semi-structured interview guide was used to explore healthcare professionals’ knowledge on PD. Most of the interviews were conducted in English, but two (2) participants requested to use both English and ‘Twi’, a local Ghanaian language. Clinical staff were approached on one-on-one basis and the study objectives were clearly explained to all participants and the voluntary nature of the study were explained and reassurance of confidentiality were also given. All quotations in the ‘Twi’ language was translated into English by a social scientist who is fluent in the “Twi” language and back translated by SFA, the first author to ensure the same meaning. Personal identifiers were removed before data collection. The interviewees were given code numbers (example Doctor D 01, Midwife M 01) to ensure anonymity. Audio files were later transferred from the recorder to the lead author’s computer and protected with a password and only accessible to the research team.

The interview guide was pretested with three (3) midwives working at the lying-in ward of the O&GD. This helped us to know the appropriateness of the guiding questions and hence there were no changes made. Data collection started from November 2021 to December 2021. This centered on the general knowledge of perinatal depression from care to management. Again, experiences on knowledge in recognizing perinatal women’s needs and providing the necessary care were explored. The interviews were conducted solely by the first author (S.F.A) a qualitative researcher with both clinical and academic experience in maternal mental health specialty. There was no prior contact with participants before the recruitment procedure. The first author, speaks and writes fluently in both English and local ‘Akan’ languages. The semi-structured interview consisted of open-ended and follow-up questions (Appendix A). Responses were further probed or re-directed where necessary to ensure a better understanding of the questions by participants in accordance with the study objectives. The date, time, and place for the interviews were scheduled to suit participants in order not to interfere with work. Five participants were permanent staff of the specialist antenatal care clinic, lying-in wards and labour wards with different levels of expertise within the O&G Directorate, and two (2) psychiatrist specialists from the psychiatry unit since the obstetricians work in consultation with psychiatrists in caring for maternal depression and two midwives also worked at the Out-Patients Department. Due to busy work schedules, participants preferred to be interviewed immediately after close of work at the consulting rooms or outside the hospital they deemed convenient to ensure privacy and comfort. The interview lasted between 25 and 40 min and with the participants’ approval, it was audio-tape recorded and transcribed verbatim. Field notes were taken during each interview process to include non-verbal signs, researcher’s reflections and the concerns raised by respondents. The interviewer is not a worker at the O&G directorate, the OPD or the psychiatry unit, so neither did she have any influence directly or indirectly on the study setting nor the participants.

### 2.5. Data Analysis

Interviews were all transcribed verbatim. Data analysis was done concurrently with data collection using inductive thematic analysis [48]. Haase’s modification of Colaizzi’s approach was used in this research [42,49]. Initially, to fully understand the intended meaning included in the transcripts, we read the transcripts several times to get the significance of the information gathered. Discussions were done simultaneously among research team to resolve differences and this helped build consensus on identified themes that needed more elaborations in subsequent interviews. Using QSR NVivo 12 (QSR International, Doncaster, Australia), significant phrases were identified and rephrased in order to transform participants’ concrete language into a language of science [50]. For example, the phrase “those who have mental illness” was restated as “facility users” and “financial inability” was changed to “financial constraints” and assigned codes to then using the “nodes” function of the software. Meanings were formulated and validation of meaning by the research team after enough discussions to reach consensus. Preliminary themes which were established, followed in subsequent interviews and then substantiated with field notes to develop themes into clusters. Again, responses from participants were summarized and confirmed with its validity with participants after each interview to determine the correctness and accuracy of the individual interviews.

#### Trustworthiness

Measures were adopted to ensure trustworthiness and credibility such as the use of the same interview guide. The in-depth interviews followed by peer debriefing ensured credibility [42]. Purposive sampling technique ensured that participants with requisite experiences on the subject of study were enrolled. To ensure conformability, member checks with three participants coupled with concurrent data analysis helped ensure the in-depth understanding and appropriate presentation of participants’ knowledge. A detailed description of the study setting, design, methodology, and participants’ background as well as enough quotations from the interview have been provided to allow for transferability in similar settings [51]. Moreover, detailed field notes were taken during each interview, and findings were discussed among authors which ensured the auditability of the study. SFA analyzed the data, and Ma Fe, confirmed the findings and discourse was discussed between both authors.

## 3. Results

### 3.1. Background Characteristics of Participants

HCPs purposely included in the study were 12, but one opted out, leaving 11 participants with an average age of 36 years old with a range of 29–51 years. Participants had been in practice for an average of 9 years. Six midwives had bachelor’s degrees, and two had diplomas in midwifery. One doctor was a senior specialist gynecologist, and the remaining two were specialist psychiatrists. All participants were Christians. Eight were currently married with one divorced. Participants with children (*n* = 8) had an average of 2.3 living kids (range = 1–3). From the interview, the time spent with patients on daily basis was mostly six to eight hours depending on one’s shift of duty. Health professionals in this study shared their experiences with their knowledge of perinatal depression. The themes that emerged were: ineffective communication lines and referral lapses, long waiting time and lack of confidentiality, poor attitude of service providers, and workload. The four main themes had sub-themes as shown in Table 1.

### 3.2. Main Findings

From the study, the general knowledge on depression during pregnancy and postnatal periods among women was described by the participants. The HCPs in their clinical practices noticed the signs and symptoms of PD in women within the early stages of antenatal and postnatal care and were able to manage. However, beyond depression, anxiety, or stress symptoms exhibited by the mother, knowledge on PD was generally low. Moreover, participants also shared their experiences on their encounter with perinatal women who experience depression in the prenatal and postnatal periods. The majority of the health professionals had also had real encounters with women who exhibited signs of depression during pregnancy or after delivery. Four main theme categories emerged from the data analysis “ineffective communication”, “poor attitude towards patients”, “workload”, and “reaction to clients symptoms”. The sub-themes that emerged also were: referral lapses, long waiting time, and lack of confidentiality during consultation, low level of knowledge, insufficient staff to meet perinatal women’s need, time constraints, identifying patients’ symptoms, assessment through patients’ centeredness, and education and counselling giving to women after recovery.

### 3.3. Theme 1—Ineffective Communication

With the first theme that emerged, HCPs admitted and explained how ineffective interpersonal relationships between themselves and perinatal women contributed to inability to easily identify and treat PD on time. This also led to patients not opening up during consulting periods. One participant also believed culture played a major role as women may not open up about their emotional problems due to cultural norms and beliefs:

“*We ask of their general well-being and about their babies but we do not include their emotional aspect. This has been the norm. It could be due to our traditional and social values. It could be that their husbands don’t treat them well but they won’t tell you unless you have your own way of getting such information and it could be cultural barriers and also stigma for those who have had mental illness in the past, they might not volunteer the information because of stigma.*”M 03

“*Well, one, they do not have the knowledge that they should seek for intervention if they have emotional problems; they do not see it as a mental problem. Then, two, their culture will tell them it is spiritual so they should pray about it and the third one is stigma*”M 06

However, one participant was of the view that stigma was the main contributing factor:

“*I will not agree with the culture aspect. I will agree to maybe stigma. Because of stigma that’s why most women don’t really talk about their problem even when asked.*”M 02

#### 3.3.1. Referral Lapses

HCPs expressed their view that there were no straight guidelines to the management of perinatal depression within the facility, which hindered PDs’ early detection and management. However, perinatal depression was managed in consultation with the psychiatrists when detected within the facility. Participants explained that when an in-patient showed signs of PD especially after delivery, a consult letter from the O&G was sent to the psychiatrist. Despite the urgency of attention the patients may need, the psychiatrists do not show up immediately sometimes:

“*When a consult letter is sent to the psychiatric unit, sometimes it takes them days before they go and see the woman and it is bad and discouraging!*”M 01

During the interview, it was gathered that lack of proper referral options and institutional barriers contributed significantly to the improper handling of perinatal depressive women:

Another midwife emphasized:

“*Most of the time, because staff at the peripheral hospitals don’t do assessment on patients properly by screening and just refer them to us, we also receive them and start treatment without screening and it becomes hard to have any suspicions of depression*”M 06

“*Some women refuse to attend antenatal clinics until their third trimester. These hospitals they attend at the last minute, will refuse to care for them and would just refer them to us without prior communication or whatsoever. This becomes frustrating as we may not have any idea about the woman’s previous medical history*”M 03

#### 3.3.2. Long Waiting Time

Participants mentioned that, some of the reasons why they preferred not asking perinatal women on their emotional well-being was due to the huge number of mothers attending the antenatal and postnatal clinics and long waiting times, that some mothers got discouraged and would seek alternate treatment elsewhere.

“*Oh yes, we are aware that when people come and there’s a queue, they will go elsewhere another time. So staying long before seeing a doctor can determine whether people should seek help*”M 04

Participants however justified why long waiting times forms part of the process for perinatal women who wait at the Out Patient Department (OPD) to be cared for:

“*Long waiting times as it stands now is inevitable. We have to see patients who had deliveries and those who underwent caesarean sections as well as those on admissions with pregnancy-related complications first, before we see those for OPDs*”D 1

“*The doctors, I believe would not like to waste any further time of their patients who may seem tired and therefore will be more concern on their physical aspect in order to see all the other women in queue*”M 08

#### 3.3.3. Lack of Confidentiality

Participants explained that lack of space in consulting rooms, which made them skip on personal and private questions especially on the woman’s emotional problems. It also prevented mothers from opening up as there was no privacy:

“*It is true and again all go down to what they have seen and how peoples’ personal information has been handled. A lot of times, two doctors are consulting in the same area and everybody can hear you and nobody is comfortable in that kind of setting. So there is no privacy*”M 04.

Participants also expressed their opinion on the fact that perinatal women do not open up because of trust issues and stigma, especially if patients’ emotional problems are family related:

“*Erm…we usually take social history of the patient, family history and then delve in more when taking such histories. At times the person knows that the mom had this very condition around this particular time but for stigmatization she is not ready to tell you that yes, my mother said she had this, she had that or it was my senior sister, no!. So the only answer she will be giving to you will be no, just for you to let her go through the process then she can then have time for herself*”M 05

During consulting, participants stated that perinatal women hardly open up unless in emergency when relatives have brought them in:

“*I believe that, even the relatives of perinatal women only open-up when she has been diagnosed of post-partum depression or when they send her to the hospital and report of depression*”D 3

### 3.4. Theme 2: Poor Attitude towards Patients

Both doctors and midwives admitted that their attitudes did not allow mothers attending clinics to communicate effectively with them regarding any peculiar psychological or problems they might have been encountering:

“*We are not ready to listen; we are ready to give instructions but we are not ready to get feedbacks. The manner in which we ask questions will give us the feedback we are looking for or the feedback we are not expecting*”M 01.

“*How you, as a midwife goes about it, how you approach the patient and how you present things to her. Approach can make her refuse*”M 02.

“*Sometimes the workload makes it difficult for you to observe your client well. Unfortunately, at times we turn a deaf ear (laugh) to the affected person*”M 06

#### 3.4.1. Low Level of Knowledge

Exceptionally in this study, all eleven HCPs knew the signs and symptoms of perinatal depression and mentioned that some of the signs and symptoms women exhibited to signify depression included being tearful, social withdrawal, lack of concentration during clinical consultation and interactions, and feeling overwhelmed with their new role as mothers or unable to accept the new role. However, low knowledge and lack of skill to manage PD was a big challenge:

“*We lack more skills and knowledge on perinatal depression*”M 01

A participant and specialist psychiatrist, who have had so many encounter with women with perinatal depression, shared his observations:

“*I think many health workers might not know the severity but those who might come with mild problems, we don’t screen them and it is as a result of lack of knowledge by health workers…*”D 01.

“*Actually, management of the Directorate usually organise workshops for us but most at times it is centered on respectful maternity care and not on perinatal mental health.*”M 08.

#### 3.4.2. No Screening Tools

Participants complained of the lack of screening tools for the detection of depression among women during pregnancy and after delivery. HCPs explained how they go about their daily routine during antenatal and postnatal clinics:

“*’We don’t routinely screen. There are no screening tools available here*’

“*’There is no particular screening tool we use. When we see signs of depression or aggression and we admit immediately after which we sedate’*”D 2

A participant also explained how he managed perinatal women with depression without screening:

“*We admit them immediately. It’s a matter of prioritizing especially in severely depressive cases. Medications like anxiolytics are served, then a consultation letter is sent to the psychiatrist.*”D 1.

### 3.5. Theme 3: Workload

According to participants, busy clinic days and overwhelming numbers of patients at particular times made them ask general questions rather than individual or private questions in order to attend to everyone on time. Emotional or psychological questions were excluded in order to avoid delays:

“*So when I am supposed to see 20 or 50 women within my shift, what am I supposed to do? Definitely, I wouldn’t touch on individual things but everything on the general. So it’s the general information that goes around without paying attention to the individual thing*”M 04.

“*I think it’s the nature of our work too. The pressure, so we are always in a hurry to care of a patient, we are not so much concerned about going into detail*”M 05.

#### 3.5.1. Insufficient Staff to Meet Perinatal Women’s Need

A participant revealed that two of their colleagues might have been trained in perinatal mental health so in their absence, it may have been difficult to manage such cases. She expressed concern for all to be trained:

“*No, I heard they have trained two (2) or three (3) people on that. So, without…Yeah so without them I don’t think generally we have anyone to cover up for them if they are not around, that is we don’t know anything about. So ideally, I think we have to train each and every one; we have to have workshop for us to know about it but it hasn’t been done yet*”M 03.

Another participant attributed the number of perinatal women who attend clinic in a day as part of the reason they did not screen for depression:

“*The number of women who come for postnatal and antenatal care outnumber us. The women we see in a day are huge, that is also another reason why*”M 07.

#### 3.5.2. Time Constraints

Limited time to attend to all perinatal women and have enough time for their complaints during antenatal and postnatal sessions was described by the participants as the main barrier that impeded effective assessment for depression detection.

“*I think we don’t get time to sit the patients down and listen to them because of the workload and pressure. We have to work on time so we don’t get enough time for one person*”M 06

Participants explained that getting detailed information from one perinatal woman to the other may be time consuming as some may be hesitant in responding; instead, they conducted their nursing care with general questions on well-being. One midwife also gave their reason as follows:

“*I think that the barriers might be time constraints with the patient because if you want to delve into a patient’s social issues and she may not be willingly giving out information you are going to spend quite some time with her whiles others will be waiting*”M 07

### 3.6. Theme 4: Reaction to Patients Detected Symptoms

Participants in this study shared their thoughts on how they reacted to initial recognitions of pregnant or post-partum women with depression. Reactions described by these health professionals ranged from being scared, burdened, and anxious to worried. A midwife who had an encounter with PD patient and a gynecologist described her feelings as follows:

“*It was terrible! The woman looked so depressed and lean during antenatal care. After asking her if there was something bothering her, she simply said no but I could sense that something was wrong! Or perhaps she might be a facility user*”M 01.

He expressed his immediate action after recognizing a patient in a severely depressive mood during the post-natal period.

An initial demonstration of confusion as to what to do upon seeing perinatal women being depressed was revealed:

“*In fact, as a doctor, the best way is to look out for risk factors though it is sometimes worrying. Some women may have had previous episodes.*”D 2.

One midwife emphasized how they reacted to patients with depressive symptoms

“*To be sincere, we sometimes become confused as to how to handle a nursing mother who suddenly becomes aggressive towards staff and baby shortly after delivery. All we do is to call for more hands*”M 07

A senior midwife and a psychiatrist also expressed their views on available treatment modalities:

“*We usually sedate them with 10 mg i.v. diazepam when we see that the woman is aggressive and then we refer to the psychiatry unit*”M 08

“*Ideally our first line for postpartum depression is that, we go for ECT (Electroconvulsive therapy). The reason is, that produces quick respond because the mother has to feed the baby*”D 2

#### 3.6.1. Identifying Patients’ Symptoms

After perceiving and recognizing patients’ symptoms through history taking and observation, participants admitted that, lack of skills in the management of perinatal depression among midwives common. Information gathered from the interview revealed that there was a gap in capacity building though they were aware of perinatal depression and the signs and symptoms that women exhibit. A participant stated:

“*Commonest one being postnatal blues and this mimic depression because one is likely to have symptoms of depression being tearful, sad, losing interest in things you had interest in even the baby one has labored and struggled in pain she may not want to attend to the baby*”D 1

“*we inform the doctor and he would refer the patient to the psychiatrist*”M 01.

“*If the patient has delusions and hallucinations, hearing voices and possibly having believe that people want to harm her, actions may change, she may refuse food and some follow the commanding voices and throw their children away….but we do our best to prevent such*”M 08

The midwives confirmed that the only training they had received was on respectful maternity care.

#### 3.6.2. Assessment through Patient’s Centeredness

Whilst healthcare providers at the O&G directorate complained of lack of skills in handling perinatal depression, one of the psychiatrists explained the steps they routinely adopt to ensure the speedy recovery of mothers with depression once they are referred to the psychiatry:

“*At the psychiatry unit, we have the biological and psychological treatment for perinatal depression. Individually, there is psychotherapy to help modify the thinking behaviors of mothers. Biological treatments using antidepressants and of course we add physical treatment like electroconvulsive therapy (ECT) which is very effective in managing depression*”D 3.

One participant said that when relatives report a patient sudden change in the house to them during antenatal or post-natal care, they take keen interest in that matter too:

“*First of all, because we know the condition we get closer to them to ask questions and this will lead you to the core point and the answers they give you will let you know she is depressed*”M 01

Another participant also expressed the precipitating factors she would look out for in perinatal women who may be experiencing depressive symptoms:

“*Erm…well I will look at their pregnancy, birth, issues surrounding pregnancy, financial and social issues and then the cultural background of the person is coming from and again whether the person has some underlying medical conditions like diabetes, hypertension or any other social issues; her dependents, whether the person is bereaved or has lost something important. These are the issues I will look at to determine the cause*”M 02

#### 3.6.3. Education and Counselling

Participants expressed how they counsel patients who experience perinatal depression after recovery and the general health education they give. Each participant shared her experience as follows:

“*Yes! So, we can have a general health education on maybe depression then we can tail it down to individual woman. We can have erm… role play, videos telling them or making them see the need for the screen*”M 04

“*Sometimes you have to counsel the patent and it depends on what the patient will tell you. If you have to involve the psychiatrist you do. Or if you can solve, it depends on what she tells you that will help you take a decision*”M 06

“*Yes, I have experience one before. So you encourage her to talk to a close relative if she is having any problem and also advise her that it is part of postnatal process and she will resolve from it soon or later. Then you encourage her to take her medication*”M 02

## 4. Discussion

This present study explored the knowledge of healthcare professionals on perinatal depression among women in a tertiary hospital in Ghana. Our study found that, psychological well-being has been noted as a basic piece of a woman’s general prospectus, yet little attention was given to it by HCPs. Diverse social components put women at more genuine peril of poor enthusiastic health than men. Therefore this study was necessitated by the fact that, there is a research gap in the knowledge on perinatal mental health among healthcare professionals in Ghana. HCPs’ understanding of women’s pregnancy intention and the extent of social support they receive may help to improve healthy behaviors during pregnancy and consequently better maternal and neonatal health outcomes [52].

Findings from the study highlight the importance of knowledge acquisition on perinatal mental health related issues as reported in previous studies [35,37]. The study found that midwives recognized the signs and symptoms of PD but needed more knowledge and skill on perinatal mental health as they were the first point of call and spent more time with women and their babies. A recent study conducted in Ghana has shown that, integration of maternal mental health is lacking in primary healthcare settings, because decentralization of mental health service delivery has mainly been credited to regional and teaching hospitals [24] hence, the site for the study was most appropriate. The implication of our finding is that, knowledge deficit and lack of skills were high leading health professionals to overlook on perinatal women with depressive symptoms. Similar findings have been reported in a related study conducted in Ireland, about how midwives reported a lack of skill in opening a discussion with women on sensitive issues, such as sexual abuse, psychosis and intimate partner violence and so adopted a selective approach to screening for perinatal mental health problems [8,53]

Our study revealed that there were other existing challenges that impeded access to screening for perinatal depression which have been identified. These barriers included lack of screening tools, poor and ineffective communication amongst healthcare providers, lack of privacy at consulting rooms in busy antenatal clinics and this findings aligns with previous studies [19,54]. All eleven (11) participants in our study had little knowledge on perinatal mental illness that resulted in referral lapses from the peripheral hospitals which ended up in under diagnosis of perinatal depression. This is consistent with previous studies that found that some midwives lack antenatal mental health knowledge, training and confidence which may lead to less willingness to engage with maternal mental health issues [24,53] and unique in this study, seven out of eight midwives had had real encounters with women with perinatal depression.

We learnt from our study that midwives have regularly been receiving training on respectful maternity care only. Although laudable, this limits their capabilities to look out for other aspect of women’s perinatal well-being such as the emotional aspects. This corroborates a study elsewhere that found that current policies to a large extent overlook the role of health professionals within obstetric settings in recognizing and supporting perinatal women with mental health issues [55]. Previous studies have identified gaps in screening [56], as tools designed in HIC may differ in functioning as majority are not culturally validated in LMICs. Although, work overload and inadequate staff were reported, there were no screening tools available for screening during routine antenatal or postnatal visits. Most of the participants, especially the midwives, had no idea about screening tools, especially the Edinburg Postnatal Depression Scale (EPDS) since it was not available in the healthcare settings. This hindered the patients’ symptoms in being early identified. The findings of this study are counterintuitive to a systematic review on ten studies conducted in five HICs where researchers observed that well-validated screening tools were available but general practitioners [GPs] were not using them [57]. The United Kingdom study reported that majority of GPs were not using the tools routinely. Notably in SSA, misconceptions, cultural beliefs, and stigma about mental conditions such as depression made perinatal women reluctant to disclose their emotional feelings and most times led to delays in their initiation of antenatal visits. This is in line with a previous study [58]. Moreover, this study finding shows that healthcare professionals were ready to learn about the use of screening tools if they had the opportunity to be trained. In a previous study, researchers found that knowledge on perinatal depression was within training modules that addressed perinatal depression (the “Thinking Healthy program”, a manual for the management of perinatal depression designed by the WHO (2015). However, the lack of a specialist workforce was a key barrier to its implementation in LMICs [59].

The study findings show that a reluctance to identify women with perinatal depression was a result of midwives’ lack of training, which should have been included in their school curricular. This is in line with previous studies in sub-Saharan Africa that reiterated that lack of capacity building as part of nurses’ and midwives’ training curricula and mental-health-related stigma were barriers to integrated maternal mental health interventions [24,60]. This shows that HCPs’ knowledge on PD and early recognition of symptoms are very important in the diagnosis of women with perinatal mental distress. Our findings show that the absence of screening tools in healthcare settings contributed to poor knowledge of PD. Consistent with previous studies [61,62], it is worth noting that while screening is essential, building patient, as well as perinatal care professionals and enhancing practice-level capacity, is critical to enhancing care among perinatal women.

In Ghana, 95% of pregnant women access antenatal services from skilled providers such as doctors, nurses or midwives, or physician assistants [63]. Although the EPDS has been validated locally [64], our health professional informants reported that long waiting times and time constraints were two major barriers to perinatal depression screening if tools were to be available, and this is consistent with a previous study [65]. Moreover, levels of coordination amongst health professionals in both obstetric settings and mental health settings were observed to be low, which led to the underreporting and undertreatment of PD. Cultural factors such as fear of disclosure, fear of antipsychotics or antidepressants, low resources among perinatal women, and lack of mental health training among health professionals also contributed to low detection and treatment rates in the tertiary hospital. This corroborates a previous study conducted in the United States that indicated that patient-provider system-level barrier prevented treatment of PD [1]. Moreover, we found that there are no succinct guidelines to the detection, treatment, and management of perinatal depression in healthcare settings. This also corroborates with a previous study [23]. Again, routine screening to determine symptoms of depression in the perinatal period is not a routine practice in almost all clinical settings in Ghana, although a study showed that there is a locally validated screening tool [64]. This finding corroborates another study [54] that screening is not performed in busy antenatal clinics. Another study, however, reiterated that screening cannot be effective without appropriate follow-up [23]. It is therefore necessary that community-based perinatal care be established for well-defined follow-up care for mothers by non-professional community health workers. This is consistent with a prior study [66] and is recommended in the WHO guidelines for informal workers working with people with mental disorders in non-specialized settings [67]. We are yet to know any previous study that have explored healthcare professionals’ knowledge on perinatal depression in Ghana. The role of nurses and midwives as primary care providers of healthcare in all communities cannot be understated. The difference in the increase in awareness and screening about the importance of perinatal mental health between HIC and LMICs is the dissemination of research and increased education among health professionals in HIC over the past decade [68].

## 5. Strengths and Limitations

Our participants had all specialised in their respective fields of work and therefore shared a precise knowledge of their experiences. Our study provides a rich in-depth explanation of how health professionals see and acknowledge symptoms of PD. However, there were limitations. The interviewer encoded the data, and the data were coded by the first author and subsequently evaluated by a single researcher. Only a small sample of health professionals was involved in the study, and moreover, perinatal women’s perspective could not also be evaluated. Again, as a result of limited financial and time resources, a guided questionnaire was considered the best and most appropriate method of data collection.

## 6. Implications for Policy and Practice

The findings reported in this study have relevant implications for clinical practice as well as to inform policy. Rigorous efforts should be put in place to equip healthcare professionals to ensure effective mental healthcare to women during the perinatal period. The study also recommends the training of obstetrician-gynecologists, midwives, mental and general nurses on the use of standardized screening tools that are locally validated for screening women for depression, anxiety, and other mental health-related disorders routinely during the perinatal period. Mental health referral system networks are also highly recommended among healthcare professionals in both primary and tertiary hospitals. It is also important to improve health professional’s awareness on perinatal mental health issues in order to support women with early symptom onset. Moreover, there is a generally an urgent need to develop protocol guidelines that will improve nursing and midwifery care plan for perinatal women. Additionally, this study calls for periodic in-service training on perinatal mental health for midwives and nurses on the assessment of psychological and emotional care during routine antenatal and postnatal care. Relatedly, it is known that midwives in Ghana avoid initiating conversations on mental issues with perinatal women because of fear that it may precipitate the thought of suicide or harming baby among women [24]. This study recommends that continuous professional development should be centered on health professionals’ mental health needs while enhancing their knowledge. National guidelines for clinical practice on perinatal mental health can be introduced to create public awareness. Increased public awareness will minimize the silence that surrounds perinatal mental issues among care providers. It is also recommended that the Government of Ghana as well as all stakeholders in the provision of healthcare in Ghana provide enough infrastructure, especially in maternity consulting rooms, to ease congestion. This would ensure privacy and make it relatively easier for patients to discuss their emotional and psychological problems with their care providers. The government should also take the necessary measures to reduce the midwife-population ratio by recruiting more midwives and nurses to minimize the workload. This will enable midwives to give enough attention to women attending antenatal and postnatal care to promote their emotional, psychological and mental well-being.

## 7. Conclusions

Our study has contributed much insight into HCPs’ knowledge on perinatal depression among women in Ghana. However, the data were collected from a small sample of subjects. Our study suggests that healthcare professionals need sufficient training on the care and management of perinatal depression among prenatal and postnatal women. Knowledge on perinatal depression, causes, and precipitating and predisposing factors as well as treatment options should be considered as part of history taking during the first antenatal and postnatal visits among perinatal women and through routine screening. To improve HCPs’ knowledge on perinatal depression, it is imperative for the hospital administrators to invest in continuous training and professional development programs for healthcare professionals. In conclusion, this study serves as a foundation for future research that will determine the learning needs of expert professional health workers on maternal mental health issues in Ghana. Stronger and more effective referral pathways that can improve access to perinatal mental healthcare in many clinical settings for women with perinatal depression are needed to improve maternal mental healthcare. Considering HCPs’ contact with women in the perinatal period, they are at an advantage for identifying women at risk, ensuring early intervention, and providing support [8].

## Figures and Tables

**Table 1 ijerph-19-15960-t001:** Themes and Sub-Themes.

Themes	Sub-Themes
Ineffective communication	Referral lapses. Long waiting time Lack of confidentiality
Poor attitude towards patients	Low level of knowledge No screening tools
Workload	Insufficient staff to meet perinatal women’s need. Time constraints
Reaction to patients symptoms	Identifying client’s symptoms Assessment through patient’s centeredness Education and counselling

## Data Availability

Due to ethical considerations, supporting data are only available through the corresponding author, S.F.A., upon reasonable request.

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
