# Peer review of "Understanding Healthcare Professionals’ Knowledge on Perinatal Depression among Women in a Tertiary Hospital in Ghana: A Qualitative Study"

_ijerph, 2022, doi:10.3390/ijerph192315960_

Round 1
Reviewer 1 Report
The topic of the manuscript is extremely necessary and important, timeless and worthy of publication. My minor comments on the manuscript-under.
The title is too long. Please remove the sentence "The body has no secrets'' and the dot at the end of the sentence.
In the section "Introduction," you write: Evidence
suggest that the most common psychological problem affecting women during the perinatal period is depression....a then...According to the World Health Organization, in high-income countries 1 in 10 women develops perinatal depression, while in low-middle-income countries 1 in 5 women suffer from perinatal depression. Perinatal depression is common and is associated with adverse effects on the mother, fetus, child and family. Depression during pregnancy has been shown to be a cause of premature birth, low birth weight, restriction of exclusive breastfeeding and stunted growth of newborns after birth...... In the literature cited, did the authors find the role of husbands/partners in offsetting the effects of perinatal depression in women? What is the positive role of men in the topic described in this article? Complete the "Conclusions" section is too laconically written. The "References"-section needs thorough revision-according to the requirements of the journal.
Regards
Author Response
Thank you very much for your comments on the Manuscript. Please, kindly find attached responses to your comments for your perusal.

Reviewer 2 Report
The introduction focuses on problems related to perinatal depression experienced by pregnant women in Ghana. But actually, the study's methodology and results of the study are designed to find out the difficulties presented by healthcare professionals in planning and developing clinical care for these patients. Perinatal depression may be present during pregnancy and in postpartum, but it was not clear which one is intended to explore; both of them are quite different in some aspects.
Despite the main objective intends to explore the knowledge of healthcare professionals about the problem of perinatal depression, the results are focused on the difficulties in caring for these patients.
In the description of the methodology, the ethical considerations do not appear and there is no reference number described by the ethics committee that has approved this study. The inclusion criteria is not clearly explained.The four thematic categories presented in the results are clear, but not some of the many subthemes described; sub-themes have a single reference quote and this may be insufficient for their justification. One of the quotes from participant M04 is almost identical to the textual section provided by participant M07.
Author Response

(The authors gave the same response as above.)

Reviewer 3 Report
The study investigates an important area of perinatal health, especially in terms of application repercussions.
Some limitations of both the research and the description given in the methodology section call for necessary additions.
INTRODUCTION
It is good to remind the reader that the EPDS is not a diagnostic tool but detects the level of symptoms and the risk of depression.
DESIGN
How many professionals were asked to participate in the study? How many professionals refused?
How many professionals were men and how many were women?
LIMITATION
It should be reported that the interviewer was also the one who encoded the data.
Furthermore, the data were coded by a single evaluator with the subsequent verification carried out by a single researcher.
Finally, it should be noted that the study sample consisted of a small number of professionals.
CONCLUSION
Point out that, in any case, the data were collected from a small number of subjects.
In light of these last considerations, I believe that the work could be placed as an exploratory study.
Author Response

(The authors gave the same response as above.)
